# Prevalence and social determinants of anxiety and depressive disorders and symptoms among adults in Ghana: A systematic review and meta-analysis

prevalence and social determinants of mental health; anxiety disorders and symptoms; depressive disorders and symptoms; sustainable development goals

**Corresponding author:**
Victoria Awortwe;
Email: victoria.awortwe@uu.se

Victoria Awortwe[1] ⬤ , Febrina Maharani[1], Meena Daivadanam[2], Samuel Adjorlolo[3], Erik MG Olsson[4], Louise von Essen[1], Vian Rajabzadeh[1] and Joanne Woodford[1]

[1]Complex Intervention Research in Health and Care, Department of Women's and Children's Health, Uppsala University, Uppsala, Sweden; [2]International Child Health and Nutrition, Department of Women's and Children's Health, Uppsala University, Uppsala, Sweden; [3]Department of Mental Health Nursing, University of Ghana, Accra, Ghana and [4]Cardiovascular Psychology, Department of Women's and Children's Health, Uppsala University, Uppsala, Sweden

## Abstract

Anxiety and depressive disorders are global health challenges, placing a significant burden on adults and healthcare systems in low- and middle-income countries (LMICs), such as Ghana. Social determinants of mental health, including poor healthcare access and poverty, may be associated with their prevalence. However, a paucity of prevalence data poses challenges for intervention planning and resource allocation. This review aimed to (1) examine the prevalence of anxiety and depressive disorders and symptoms among adults in Ghana, and (2) explore social determinants of mental health potentially associated with anxiety and depressive disorders and symptoms. We searched electronic databases and secondary sources from inception until September 30, 2024. Meta-analyses were performed to estimate the pooled prevalence. Narrative synthesis explored social determinants potentially associated with anxiety and depressive disorders and symptoms.

We included 38 studies (22,587 adults). Pooled point prevalence of anxiety and depressive disorders and symptoms was 40.3% (95% confidence interval [CI]: 31.8–49.4%) and 33.0% (95% CI: 27.7–38.8%), respectively. Most studies (37 studies) reported the prevalence of symptoms and not disorders. Social determinants of mental health, including educational attainment and urban environment, were associated with higher levels of anxiety symptoms, while ethnicity and traumatic experiences were associated with higher levels of depressive symptoms. There was a high degree of heterogeneity, and the majority of studies used self-report screening tools, which may have skewed prevalence estimates. More than a third of adults in Ghana were found to experience anxiety and depressive symptoms, and social determinants of mental health may be associated with prevalence. High-quality research and contextually appropriate interventions targeting the identified social determinants of mental health associated with anxiety and depressive symptoms are needed to reduce disparities and improve the mental well-being of adults in Ghana.

## Impact statement

The absence of comprehensive population-based prevalence data remains a significant barrier to informed mental health policy planning and equitable service delivery in Ghana. Our review addresses this gap and offers the first comprehensive synthesis of evidence on the prevalence and social determinants of anxiety and depressive disorders and symptoms among adults in Ghana. We found that over a third of adults may experience anxiety and depressive symptoms. We also identified a range of social determinants of mental health, including demographic, economic, neighbourhood, social and cultural factors associated with prevalence. These findings align with the sustainable development goals and hold implications for the development of both individual-level and population-level interventions. The review also underscores the urgent need to standardise and validate mental health assessment tools, a methodological challenge that extends beyond Ghana to many LMICs. Our findings also contribute to the global discourse on the need for equity-oriented mental health strategies and emphasise the value of locally grounded, yet globally relevant research and practice.





## Introduction

Anxiety and depressive disorders among adults are growing public health challenges that significantly contribute to the global burden of disease and disability (Ferrari et al., 2022).

Anxiety and depressive disorders adversely impact physical health, quality of life and pose substantial challenges to economic productivity, healthcare systems and societal development (Campion et al., 2022). The burden of anxiety and depressive disorders is particularly pronounced in low- and middle-income countries (LMICs) due to limited financial and human resources within mental health systems (Rathod et al., 2017). Examining the prevalence of anxiety and depressive disorders and symptoms, and social determinants of mental health (SDOMH) associated with their occurrence in LMICs such as Ghana, can guide interventions and policy planning.

Recent systematic reviews have found the point prevalence of anxiety and depression symptoms to be 47% and 48%, respectively (Bello et al., 2022), and lifetime prevalence to range from 3.3% to 9.8% for anxiety disorders, and 5.7% to 15.8% for depressive disorders across Africa (Greene et al., 2021; Bello et al., 2022). In sub-Saharan Africa (SSA), the burden of mental disorders is expected to increase by 130% between 2010 and 2050 (Charlson et al., 2014). This anticipated increase may pose significant challenges for countries such as Ghana, where economic constraints, limited healthcare infrastructure and a shortage of healthcare professionals already hinder the effective treatment of mental disorders. Therefore, interest-holders (*e.g.*, healthcare providers, non-profit organisations, patients, policymakers and researchers) are urging action to address the growing mental health crisis in SSA, including implementing national mental health policies, integrating mental health into primary healthcare, promoting self-care and training formal (nurses) and informal (faith healers and traditional healers) care providers (Patel et al., 2018; Aguwa et al., 2023; Sorsdahl et al., 2023). Following the staging model of mental health (Patel et al., 2018), it is important to complement these actions with prevention strategies, targeting SDOMH.

Biological, psychological and social factors influence the onset and progression of anxiety and depressive disorders (Patel et al., 2018). Traditionally, biological and psychological factors have informed psychological and pharmacological intervention development. However, even with optimal access to evidence-based interventions, an estimated 60% of the burden of mental disorders cannot be averted by such interventions alone (Andrews et al., 2004; Grummitt et al., 2023). Thus, there is a growing emphasis on reducing the burden of mental disorders *via* SDOMH that may be associated with prevalence and inequities in intervention access (Alegría et al., 2018; Lund et al., 2018; Patel et al., 2018). SDOMH are largely modifiable, and their impact can be mitigated or leveraged through targeted and universal strategies (*e.g.*, community-based programmes and social policies), to reduce the burden of mental disorders (Rathod et al., 2017; Kirkbride et al., 2024).

This systematic review focuses on Ghana, a lower-middle-income country with a three-tiered health system that delivers primary, secondary and tertiary care through public, private and faith-based providers (Assan et al., 2018). Over the past two decades, the Government of Ghana has implemented major health and social protection initiatives, including the National Health Insurance Scheme and Livelihood Empowerment Against Poverty programme, to alleviate poverty and improve healthcare access (Otieno et al., 2022). Mental health has also gained increasing policy attention, particularly with the enactment of the Mental Health Act 846 in 2012, which established the Mental Health Authority to coordinate mental health services and protect the rights of people with mental disorders (Walker and Osei, 2017). The 12-year mental health policy 2019–2030, launched in 2019, further aims to promote mental health, prevent mental disorders and strengthen the

management and care of people with mental disorders (Government of Ghana, 2018; Walker and Osei, 2017). Despite these advances, the mental health system in Ghana remains under-resourced and overstretched, with limited funding, workforce shortages, a high treatment gap and severe shortages of psychotropic medication (Asamani et al., 2021; Government of Ghana, 2018). The country also performs poorly on indicators of population well-being, such as educational attainment, environmental pollution, gender equality and multidimensional poverty (United Nations Development Programme, 2020).

Ghana is currently experiencing an epidemiological transition, characterised by a double burden of communicable and non-communicable diseases (Government of Ghana, 2018; Otieno et al., 2022). The World Health Organization (WHO) estimates that about13% of Ghanaians live with a mental disorder (Oppong et al., 2016; World Health Organization, 2021). In recent times, the country has faced macroeconomic crises, including a debt crisis, fuel shortages and rising unemployment, which may have increased the risk of mental disorders, including anxiety and depressive disorders, partly by worsening socioeconomic vulnerabilities and amplifying the adverse effects of SDOMH on mental health outcomes (The World Bank Group, 2024). While Ghana is working towards improving mental health care *via* its 12-year mental health policy (Ae-Ngibise et al., 2023; Government of Ghana, 2018; Mwangi et al., 2023), the lack of comprehensive national and local prevalence data for adult mental disorders and insufficient evidence on associated SDOMH hinders efforts to estimate the burden of disease and develop evidence-based multisectoral strategies aimed at improving mental health and well-being of adults in Ghana (Government of Ghana, 2018). Therefore, the primary aims of this review were to (1) examine the prevalence of anxiety and depressive disorders and symptoms among adults in Ghana, and (2) explore SDOMH potentially associated with anxiety and depressive disorders and symptoms. The secondary aim was to examine the prevalence of symptoms of psychological distress among adults in Ghana.

## Methods

We followed the Joanna Briggs Institute (JBI) method for systematic reviews of prevalence (Munn et al., 2015) and the Preferred Reporting Items for Systematic Reviews and Meta-Analyses (PRISMA) guidelines; see Supplementary Appendix A (Page et al., 2021). We published the review protocol (Awortwe et al., 2024) and it has been registered in the International Prospective Register of Systematic Reviews (PROSPERO, CRD42023463078). Protocol changes made before analysis are recorded in PROSPERO and reported in Supplementary Appendix B.

## Search strategy

We searched nine electronic databases (Supplementary Appendix C) from inception up to September 30, 2024. We manually reviewed the reference lists of included studies and identified systematic reviews by conducting forward citation checks of included studies. We searched grey literature in the Agency for Healthcare Research and Quality, Google Scholar, Health Systems Trust, Open Grey (http://www.open grey.eu/) and the WHO websites. We contacted researchers and non-governmental organisations working in adult mental health in Ghana to identify unpublished or ongoing studies.

We constructed the search using a combination of Medical Subject Headings and free text words for "mental disorders," "anxiety disorders and symptoms," "depressive disorders and symptoms" and "Ghana," with assistance from a librarian at Uppsala University. The electronic search strategy was peer reviewed by two researchers following the PRESS guidelines (McGowan et al., 2016). The PubMed search strategy is shown in Supplementary Appendix C. Search strategies for all electronic databases and the PRESS peer review have been published (Awortwe et al., 2024). Only studies available in English and Ghanaian languages (e.g., Ewe, Ga and Twi) were eligible.

## Eligibility criteria

Eligibility criteria follow CoCoPops (Condition, Context, Population and study design) format:

### Condition

Eligible studies assessed anxiety disorders (e.g., generalised anxiety disorder and phobias) and depressive disorders using (1) a structured diagnostic clinical interview following the International Classification of Diseases and Related Health Problems or the Diagnostic and Statistical Manual of Mental disorders; and/or (2) anxiety and depressive symptoms using a validated self-report, clinician or proxy-administered screening tool for anxiety, for example, Beck Anxiety Inventory (BAI) (Beck et al., 1988), and depression, for example, Beck Depression Inventory (BDI) (Beck et al., 1961) in an adult sample. There was no restriction on cutoff scores for self-report screening tools due to research suggesting that commonly used standardised cut-offs may not be suitable for adult populations in LMICs (Ali et al., 2016; Mughal et al., 2020). Studies reporting point, period or lifetime prevalence of anxiety and depressive disorders and symptoms were eligible. Given symptoms of psychological distress encompass symptoms of common mental disorders, such as anxiety and depression in adults (Duarte and Jiménez-Molina, 2022), the prevalence of symptoms of psychological distress was examined as a secondary outcome. Studies using validated measures, for example, Kessler Psychological Distress Scale (K10) (Kessler et al., 2002), to report the prevalence of symptoms of psychological distress in adult samples, were eligible. We excluded studies if prevalence could not be calculated (i.e., symptoms reported only as mean scores) and/or obtained from the study authors.

### Context

Eligible studies were conducted in (1) Ghana with adults sampled from the community and/or clinical settings including through online recruitment, or (2) regions that encompass Ghana (e.g., West Africa and SSA) with data on participants living in Ghana that could be extracted or obtained from the study authors. Studies conducted immediately (i.e., <4 months after the official end date) of ethnic and/or political conflicts (e.g., Konkomba-Nanumba conflict and Western Togoland crisis), humanitarian crises or natural disasters were excluded (Lim et al., 2022).

### Population

Eligible studies included adults (aged ≥18 years). Studies including participants aged <18 years were excluded if they did not present data separately for adults, or if this data could not be obtained via correspondence with study authors. Informed by similar systematic reviews conducted in SSA (Jörns-Presentati et al., 2021; Bello et al., 2022; Kaggwa et al., 2022; Opio et al., 2022), studies including adults with comorbid physical health conditions were eligible (Austin, 2024; Bhattacharya et al., 2014; Centers for Medicare & Medicaid Services, 2020). Studies focusing on other mental disorders (e.g., personality disorders and psychotic disorders), substance use (e.g., alcohol dependence) and neurological disorders (e.g., multiple sclerosis) were excluded. Studies on prisoners and adults accused of witchcraft were also excluded because these adults are not necessarily representative of the wider Ghanaian population.

### Study design

Eligible studies were quantitative and used observational designs, such as longitudinal cohort (baseline data only), case–control and cross-sectional studies. Mixed methods studies were eligible if quantitative data could be extracted or obtained from the study authors. In studies conducted on the same cohort of individuals at the same or different time points, or where samples overlapped, the study with the largest sample and primary outcomes related to this review was included to ensure duplicate data were not included (Opio et al., 2022). Studies reporting secondary data analyses were excluded. Published and unpublished studies were eligible for inclusion.

## Study selection

We de-duplicated searches using EndNote V.20 (Clarivate) and imported them into Rayyan for study screening and selection. Title and abstract screening, followed by full paper checks of potentially eligible studies, were conducted independently by two reviewers (F.M. and V.A.). Disagreements were resolved through discussion or consulting a third reviewer (J.W.). Authors were contacted a maximum of two times via email for additional information to determine study eligibility, where necessary. We excluded studies if authors did not respond or provide the requested information.

## Data extraction

Data were extracted independently by one reviewer (V.A.) into a standardised Microsoft Excel form (see review protocol) and cross-checked by a second reviewer (F.M. or V.R.).

## Quality assessment

Two reviewers (V.A. and F.M.) independently conducted the quality assessment of the included studies using the JBI Critical Appraisal Tool for Prevalence Studies (Munn et al., 2015). Each domain was rated as Yes, No, Unclear or Not applicable (Munn et al., 2015), with reasons for ratings provided in Supplementary Appendices D and E. The total score ranged from 1 to 9 for individual studies, with the total number of "Yes" scores for individual studies used to appraise studies as low (≤3 points), moderate (4–6 points) or high-quality (≥7 points). Disagreements were resolved by discussion between the two reviewers or by consulting a third reviewer (V.R. or J.W.).

## Meta-analysis

We conducted the meta-analysis using Comprehensive Meta-Analysis software V.4. (Borenstein, 2022). Individual study data on the point prevalence (i.e., number of cases/sample size) of

anxiety disorders and symptoms, depressive disorders and symptoms and symptoms of psychological distress, respectively, were extracted. Data were transformed to their logits to stabilise variances before performing the meta-analysis (Doi and Williams, 2013; Yang et al., 2023). We adopted a random-effects model given anticipated heterogeneity across studies. We assessed heterogeneity using Cochran's $Q$ statistic and measured the proportion of total variability due to between-study heterogeneity using the $I^2$ statistic. The prediction interval ($T^2$) was used as an estimate of between-study variance in true effects observed in eligible studies. We performed a sensitivity analysis excluding studies (1) conducted during coronavirus disease 2019 (COVID-19) pandemic, and (2) of lower quality (*i.e.*, quality score ≤ 3), from the analysis to examine whether removal caused any substantial change to overall pooled prevalence estimates. We assessed publication bias using Egger's regression statistic and funnel plots (Sutton et al., 2000) and Duval and Tweedie's trim-and-fill method to identify and correct for funnel plot asymmetry arising from publication bias (Duval and Tweedie, 2000).

## Subgroup analysis

We conducted a subgroup analysis to explore sources of heterogeneity and examine the potentially moderating effects of the following factors on prevalence:

- Presence of a chronic physical condition (yes/no)
- Type of chronic physical conditions (*e.g.*, cancer, diabetes and human immunodeficiency virus [HIV]/acquired immunodeficiency syndrome [AIDS])
- Method of assessment (structured clinical interview or screening tool)
- Sample size (<384 *vs.* ≥384)
- Study design (case–control study, cross-sectional studies and longitudinal cohort study)
- Study quality (low, moderate or high-quality studies)
- Time period of data collection (<2020 *vs.* ≥2020).
- Population type (general population *vs.* specific population). The term "general population" describes adults without distinguishing characteristics related to age, occupation, physical condition or comorbidity, whereas "specific population" describes adults with specific attributes, for example, chronic physical conditions, caregivers, employees, students and older adults (Kaggwa et al., 2022).

Statistical significance level for subgroup analysis was set at $P < 0.05$ (two-tailed).

## Narrative synthesis

SDOMH potentially associated with anxiety and depressive disorders and symptoms were narratively synthesised, with proximal and distal factors categorised under the demographic, economic, neighbourhood, environmental events, social and cultural domains of the social determinants of mental disorders and sustainable development goals (SDGs) framework (Lund et al., 2018).

## Results

### Study selection

We identified 4,934 records *via* electronic database searches and other methods. We assessed 272 full-text records for eligibility, of

which 38 studies met the inclusion criteria (see Figure 1 and also Supplementary Appendices G and H for detailed reasons for study exclusion).

### Study characteristics and quality

Study characteristics are reported in Table 1. We included 38 studies ($N$ = 22,587 participants – 9,783 males, 12,103 females and 701 whose sex was not specified). Mean age ranged from 21.3 (±2.4) to 69.9 (±8.8) years, and the sample size ranged from 38 to 5,391. Study samples included general populations (7 studies), specific populations (29 studies) and mixed population groups (2 studies). Studies were conducted in community (13 studies), hospital (19 studies), mixed community and hospital (1 study), and online (5 studies) settings. Most studies used cross-sectional designs (35 studies) and were published between 2013 and 2024.

Data on the point prevalence of anxiety disorders and symptoms were reported in 19 studies ($N$ = 7,705). One study assessed the prevalence of agoraphobia, generalised anxiety disorder and social phobia, using the Mini International Neuropsychiatric Interview (MINI) (Sheehan et al., 1998). The remaining 18 studies assessed the prevalence of anxiety symptoms using self-report screening tools, including the BAI (2 studies), the anxiety subscale of the Depression Anxiety and Stress Scale (DASS-21; 9 studies) (Lovibond and Lovibond, 1995), the General Anxiety Disorder Scale (GAD-7; 2 studies) (Spitzer et al., 1999), the anxiety subscale of the Hospital Anxiety and Depression Scale (HADS; three studies) (Zigmond and Snaith, 1983) and the State–Trait Anxiety Inventory (STAI; two studies) (Spielberger, 2010).

Data on the point prevalence of depressive disorders and symptoms were reported in 30 studies ($N$ = 12,950). One study assessed the prevalence of major depressive disorder using the MINI. The remaining 29 studies assessed the prevalence of depression symptoms using self-report screening tools, including the first edition of the BDI (3 studies) (Beck et al., 1961), second edition of the BDI-II (1 study) (Beck et al., 1996), Center for Epidemiological Studies-Depression Scale (CESD-10; 2 studies) (Radloff, 1977), the depression subscale of DASS-21 (9 studies), the Geriatric Depression Scale (GDS-15; 2 studies) (Yesavage and Sheikh, 1986), the depression subscale of the HADS (4 studies) (Zigmond and Snaith, 1983), Patient Health Questionnaire (PHQ-9; 7 studies) (Kroenke et al., 2001), and the Zung Self-Rating Depression scale (1 study) (Zung, 1965).

Data on the point prevalence of symptoms of psychological distress were reported in 6 studies ($N$ = 8,660). Two instruments, comprising the K10 (5 studies) and the PHQ for Depression and Anxiety (PHQ-4; 1 study) (Kroenke et al., 2009), were used to assess symptoms of psychological distress.

Quality assessment scores ranged from 1 to 9; most studies were of moderate quality (65.8%, 25 studies, see Supplementary Appendix E). High risk of bias was most frequently identified in domains related to appropriateness of sampling methods (25 studies), adequateness of sample size (14 studies), standard and reliable measurement of condition (28 studies), sufficient coverage of identified sample (16 studies) and, adequateness and appropriate management of response rate (25 studies).

### Prevalence of anxiety and depressive disorders and symptoms

The pooled point prevalence of anxiety disorders and symptoms was 40.3% (95% confidence interval [CI]: 31.8–49.4%, $I^2$ = 98.0%, $\tau^2$ = 0.642, $T^2$ = 10.6–79.3%; Figure 2). The pooled point prevalence

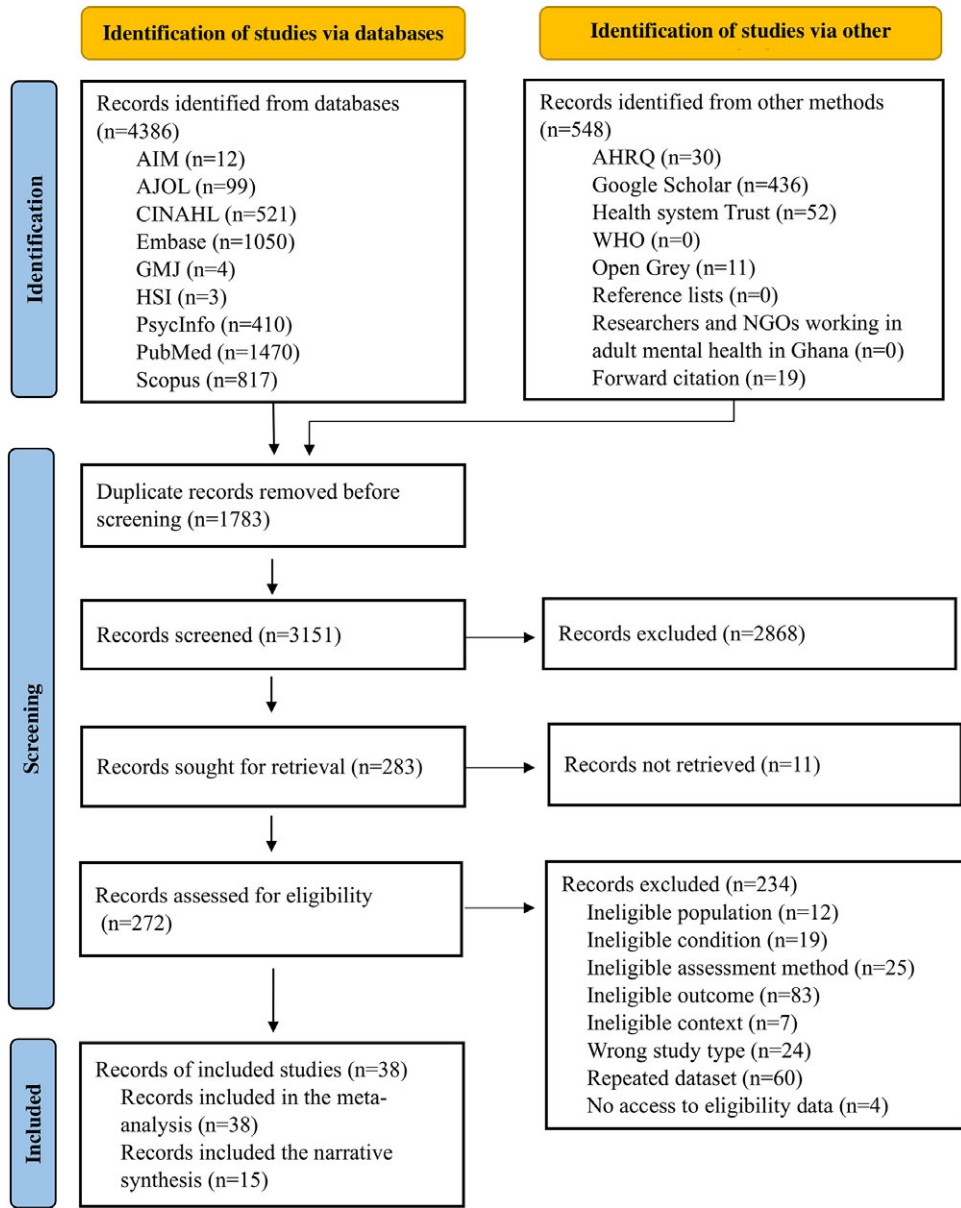

**Figure 1.** PRISMA flow diagram of study selection.

of depressive disorders and symptoms was 33.0% (95% CI: 27.7–38.8%, $I^2$ = 97.0%, $\tau^2$ = 0.461, $T^2$ = 10.7–67.0%; Figure 3).

### Sensitivity analysis and publication bias

Sensitivity analysis revealed no significant change to the pooled point prevalence of anxiety disorders and symptoms, depressive disorders and symptoms, and symptoms of psychological distress, respectively, after removing studies conducted during COVID-19 pandemic, and studies of lower quality (*i.e.*, quality score ≤ 3) from the analysis *via* the leave-one-out method.

There was no evidence of publication bias for anxiety disorders and symptoms (Egger's test, $t$ = 1.42, $p$ = 0.173), depressive disorders and symptoms (Egger's test, $t$ = 0.83, $p$ = 0.415) and psychological distress (Egger's test: $t$ = 1.66, $p$ = 0.171). The Duval and Tweedie trim-and-fill analysis suggested imputation of one study to the left of the mean for depression, yielding an adjusted pooled point prevalence estimate of 31.6% (95% CI: 26.3–37.5%).

The pooled point prevalence of symptoms of psychological distress was 36.7% (95% CI: 25.2–50.0%, $I^2$ = 99.0%, $\tau^2$ = 0.456, $T^2$ = 7.1–81.5%; see Supplementary Appendix J). The Duval and Tweedie trim-and-fill analysis also suggested imputation of three studies to the left of the mean for symptoms of psychological distress, yielding an adjusted pooled point prevalence of 25.2% (95% CI: 16.4–36.7%, see Supplementary Appendix I–K for funnel plots).

### Subgroup analyses

#### Subgroup analysis of the prevalence of anxiety disorders and symptoms

Subgroup analysis showed that the presence of a chronic physical condition ($Q$ = 7.118, $p$ < 0.05), type of chronic physical condition ($Q$ = 9.926, $p$ < 0.05), region ($Q$ = 39.553, $p$ < 0.05) and time period of data collection ($Q$ = 16.568, $p$ < 0.05) were significant moderators of anxiety prevalence (see Supplementary Appendix L). Specifically,

**Table 1.** Study characteristics and quality assessment

| Author | Region (s) | Recruitment setting (s) | Study design | Participant | Gender | Mean age | Condition | Assessment method | Sample size | Cases | Proportion | Overall quality rating |
|---|---|---|---|---|---|---|---|---|---|---|---|---|
| Canavan et al. (2013) | Ashanti, Brong Ahafo, Central, Eastern, Greater Accra, Northern, Upper East, Upper West, Volta, Western | Households | Cross-sectional | General population | F: 2,897 M: 2,494 | 42.5 ± 0.24 | Symptoms of psychological distress | K10 ≥ 25 | 5,391 | 1,117 | 20.7 | High |
| Kretchy et al. (2014) | Ashanti Greater Accra | Tertiary hospital | Cross-sectional | Hypertensive patients | F: 251 M: 149 | NR | Anxiety symptoms Depressive symptoms | DASS–21[a] | 400 | 271 42 | 67.7 10.5 | High |
| Asante and Andoh-Arthur (2015) | Greater Accra | University | Cross-sectional | University students | F: 138 M: 132 | 22 ± 2.39 | Depressive symptoms | CESD >10 | 270 | 106 | 39.2 | Low |
| Ganu et al. (2018) | Greater Accra | Tertiary hospital | Cross-sectional | Kidney patients on haemodialysis | F: 43 M: 63 | 48.7 ± 1.33 | Depressive symptoms | PHQ–9 ≥ 10 | 106 | 47 | 44.3 | Mod |
| Akpalu et al. (2018) | Greater Accra | Tertiary hospital | Cross-sectional | Type–2 diabetes patients | F: 314 M: 86 | 52.7 ± 8.7 | Depressive symptoms | PHQ–9[a] | 400 | 125 | 31.3 | Mod |
| Gyasi et al. (2019) | Ashanti | Rural and urban communities | Cross-sectional | Older adults | F: 759 M: 441 | NR | Symptoms of psychological distress | K–10 ≥ 20 | 1,200 | 544 | 45.3 | Mod |
| Kugbey et al. (2018) | Volta | Community | Cross-sectional | Older adults | F: 161 M: 101 | NR | Depressive symptoms | GDS–15 ≥ 5 | 262 | 99 | 37.8 | High |
| Ademola et al. (2019) | Greater Accra | Tertiary hospital | Cross-sectional | Hypertensive Patients | F: 70 M: 50 | 57.0 ± 13.7 | Depressive symptoms | PHQ–9 > 4 | 120 | 50 | 41.7 | Mod |
| Awuah et al. (2019) | Ashanti | Rural and urban communities | Cross-sectional | General population | F: 1,364 M: 657 | F: 46.3 ± 12.5 M: 47.4 ± 13.2 | Depressive symptoms | PHQ–9 > 4 | 2021 | 601 | 29.7 | Mod |
| Ofori-Atta et al. (2019) | Greater Accra Ashanti | Tertiary hospital | Cross-sectional baseline | Caregivers of children living with HIV | F: 359 M: 82 NR:5 | 42.16 ± 10.44 | Depressive symptoms | BDI–21 ≥ 10 | 446 | 126 | 28.3 | Mod |
| Nuvey et al. (2020) | Eastern Northern | Communities | Cross-sectional | Cattle Farmers | F: 19 M: 268 | 46.9 ± 11.7 | Anxiety symptoms Depressive symptoms | DASS–21 | 287 | NR | 66.0 72.0 | Mod |
| Adu et al. (2021) | Ashanti Greater Accra Others | Online platforms | Cross-sectional | General population | NR: 495 | NR | Depressive symptoms | PHQ–9 ≥ 10 | 495 | 61 | 12.3 | Mod |

(*Continued*)

*Cambridge Prisms: Global Mental Health*

| Author | Region (s) | Recruitment setting (s) | Study design | Participant | Gender | Mean age | Condition | Assessment method | Sample size | Cases | Proportion | Overall quality rating |
|---|---|---|---|---|---|---|---|---|---|---|---|---|
| Amoako et al. (2021) | Central and Ashanti region | Clinic and rural communities | Case–control | Caregivers and the general population | F: 42 M: 32 | NR | Anxiety symptoms Depressive symptoms | HADS ≥8 | 74 | 11 4 | 14.9 .4 | Mod |
| Amu et al. (2021) | Oti Volta | Communities | Cross-sectional | General population | F: 1383 M: 1073 | 41.8 ± 0.3 | Anxiety symptoms Depressive symptoms Comorbid anxiety and depressive symptoms | DASS–21[a] | 2,456 | NR | 53.3 25.2 24.2 | High |
| Asare-Doku et al. (2021) | Multiple locations | Mining sites-Communities | Cross-sectional | Employees | F: 153 M: 1012 | NR | Symptoms of psychological distress s | K10 ≥ 16 | 1,165 | 436 | 37.4 | Mod |
| Boateng et al. (2021) | Multi-country | Online platforms | Cross-sectional | General population | F: 406 M: 405 | NR | Anxiety symptoms | GAD–7 ≥ 5 | 811 | NR | 11.6 | Mod |
| Ofori et al. (2021) | Ashanti | Primary healthcare centres | Cross-sectional | Healthcare workers | F: 121 M: 115 NR: 36 | 30.2 ± 5.2 | Anxiety symptoms Depressive symptoms | DASS–21-A ≥ 8 DASS21-D ≥ 10 | 270 271 | 75 57 | 27.8 21.1 | Mod |
| Saah et al. (2021) | Greater Accra | Restaurants | Cross-sectional | Waiters | F: 267 M: 117 | 23.03 ± 3.8 | Anxiety symptoms Depressive symptoms Comorbid anxiety and depressive symptoms | DASS–21-A > 9 DASS–21-D > 13 | 384 | 201 147 38 | 52.3 38.3 9.9 | High |
| Shaikh et al. (2021) | Multiple locations | Online platforms | Cross-sectional | University students | NR: 110 | 22.3 ± 3.1 | Anxiety symptoms Depressive symptoms | DASS–21[a] | 110 | 13 26 | 11.8 23.6 | Low |
| Swaray et al. (2021) | Multiple locations | Online platforms | Cross-sectional | Medical laboratory professionals | F: 95 M: 378 | 33.3 ± 6.5 | Anxiety symptoms Depressive symptoms | DASS21-A ≥ 8 DASS21-D ≥ 10 | 473 | NR | 17.8 9.1 | Mod |
| Adjepong et al. (2022) | Ashanti | Online platform | Cross-sectional | University students | F: 57 M: 72 | 21.3 ± 2.4 | Anxiety symptoms | GAD–7 ≥ 10 | 129 | 29 | 22.5 | Mod |
| Arthur-Mensah et al. (2022) | Greater Accra | Regional hospital | Cross-sectional | Healthcare workers | F: 16 M: 22 | 35 ± 8.39 | Anxiety symptoms Depressive symptoms | STAI BDI-II 21 ≥ 15 | 38 | 18.5 23 | 48.7 60.5 | High |
| Boima et al. (2015) | Greater Accra | Tertiary hospital | Cross-sectional | Male hypertension patients | M: 358 | 56.2± 13.5 | Symptoms of psychological distress | K10 ≥ 20 | 358 | 202 | 56.4 | Mod |
| Danquah and Mante (2022) | Ashanti | University hospital | Cross-sectional | COVID–19 survivors | F: 175 M: 193 | 48.58 ± 19.6 | Anxiety symptoms Depressive symptoms | STAI ≥ 40 | 368 | 169.464 158 | 46.05 42.93 | Mod |
| Omuojine et al. (2022) | Ashanti | Tertiary hospital | Cross-sectional | People living with HIV/AIDS | F: 375 M: 120 | NR | Anxiety symptoms Depressive symptoms | HADS-A ≥ 8 HADS-D ≥ 8 | 495 | 302 275 | 61.0 55.6 | Mod |

*(Continued)*

**Table 1.** (*Continued*)

| Author | Region (s) | Recruitment setting (s) | Study design | Participant | Gender | Mean age | Condition | Assessment method | Sample size | Cases | Proportion | Overall quality rating |
|---|---|---|---|---|---|---|---|---|---|---|---|---|
| Opoku Agyemang et al. (2022a) | Central Greater Accra | Tertiary hospital | Cross-sectional | Psychiatric nurses | F: 187 M: 124 | 42.5 ± 0.24 | Anxiety symptoms Depressive symptoms | BAI–21 ≥ 8 BDI–21 ≥ 10 | 311 | 84 61 | 27 19.6 | High |
| Opoku Agyemang et al. (2022b) | Central | Tertiary hospital | Cross-sectional | Outpatients living with HIV/AIDS | F: 298 M: 97 | 40.67 ± 12.5 | Anxiety symptoms Depressive symptoms | DASS–21-A ≥ 4 DASS–21-D ≥ 5 | 395 | 161 113 | 40.8 28.6 | High |
| Ae-Ngibise et al. (2023) | Ahafo Upper-East Volta | Primary healthcare centres | Cross-sectionals | General population | F: 741 M: 168 | NR | Depressive symptoms | PHQ–9 ≥ 5 | 909 | 142 | 15.6 | High |
| Agyekum et al. (2023) | Bono | Regional hospital | Case–control | Type–2 diabetes patients and healthy controls | F: 230 M: 130 | 22 ± 7.9 48.6 ± 10.6 | Depressive symptoms | PHQ–9 > 9 | 360 | 80 | 22.2 | Mod |
| Anum et al. (2023) | Greater Accra | Motor transport and Traffic Directorate units | Cross-sectional | Police officers | F: 42 M: 131 | 35.76 ± 7.52 | Anxiety symptoms Depressive symptoms | DASS–21[a] | 173 | NR | 49.1 39.8 | Low |
| Assefa et al. (2023) | Bono East | Households | Cross-sectional | General population | F: 160 M: 141 | NR | Symptoms of psychological distress | PHQ–4 ≥ 3 | 301 | 77 | 25.6 | Mod |
| Dordoye et al. (2023) | Volta | University | Cross-sectional | Medical students | F: 86 M: 146 | NR | Depressive symptoms | BDI–21 ≥ 14 | 232 | 161 | 69.4 | Mod |
| Nakua et al. (2023) | Ashanti | Households | Cross-sectional | Older adults | F: 284 M: 134 | 69.9 ± 8.8 | Depressive symptoms | GDS–15 ≥ 6 | 418 | 172 | 41.1 | High |
| Siakwa et al. (2015) | Central | Regional hospital | Cross-sectional | Outpatients living with HIV | F: 120 M:86 | 32.2 ± 12 | Anxiety disorders (agoraphobia, general anxiety disorder and social phobia) Depressive disorder | MINI | 206 | 63 71 | 30.6 34.5 | Mod |
| Kugbey (2022) | Greater Accra | Tertiary hospital | Cross-sectional | Breast cancer patients | F: 205 | NR | Anxiety symptoms Depressive symptoms | HADS ≥8 | 205 | NR | 48.5 37.3 | Mod |
| Calys-Tagoe et al. (2017) | Greater Accra | Tertiary hospital | Cross-sectional | Breast cancer patients | F: 120 | 50.3 ± 10.9 | Anxiety symptoms Depressive symptoms | HADS ≥8 | 120 | NR | 92.5 84.2 | Mod |
| Fofie et al. (2023) | Western | Primary healthcare centres | Cross-sectional | Frontline healthcare workers | F: 140 M: 105 | NR | Symptoms of psychological distress | K10[a] | 245 | 98 | 40 | Mod |
| Nutor et al. (2024) | Volta | Teaching hospital | Cross-sectional | Outpatients living with HIV | F: 25 M: 101 NR: 55 | NR | Depressive symptoms | CESD ≥16 | 181 | 37 | 20.4 | Mod |

*Note:* Overall quality rating: Low quality, Mod = moderate quality, high quality.
Abbreviations: NR, not reported; F, females; M, males.
[a]Cut-offs not reported.

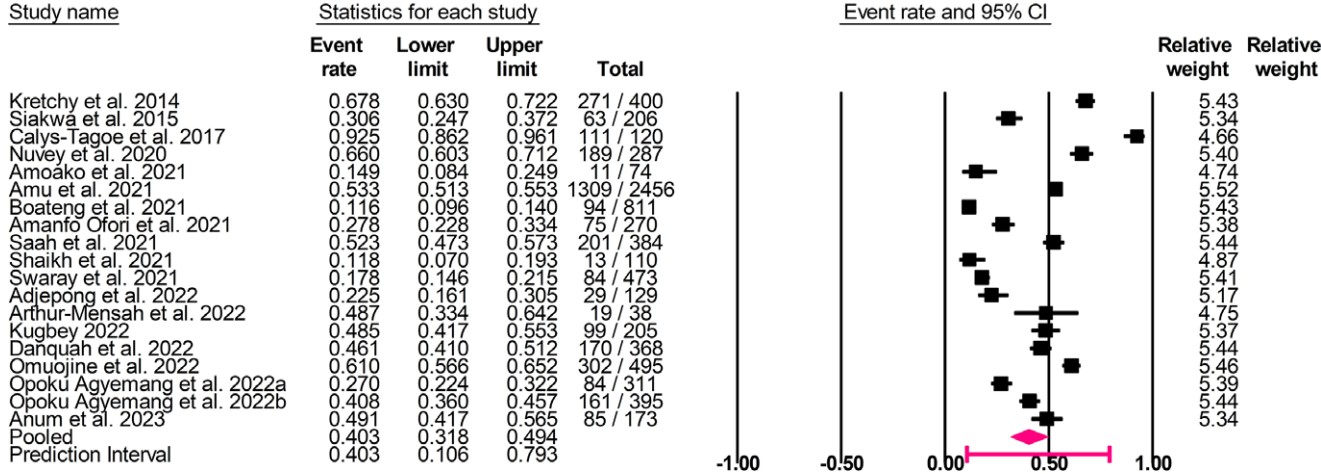

**Figure 2.** Forest plot of the pooled prevalence of anxiety disorders and symptoms.

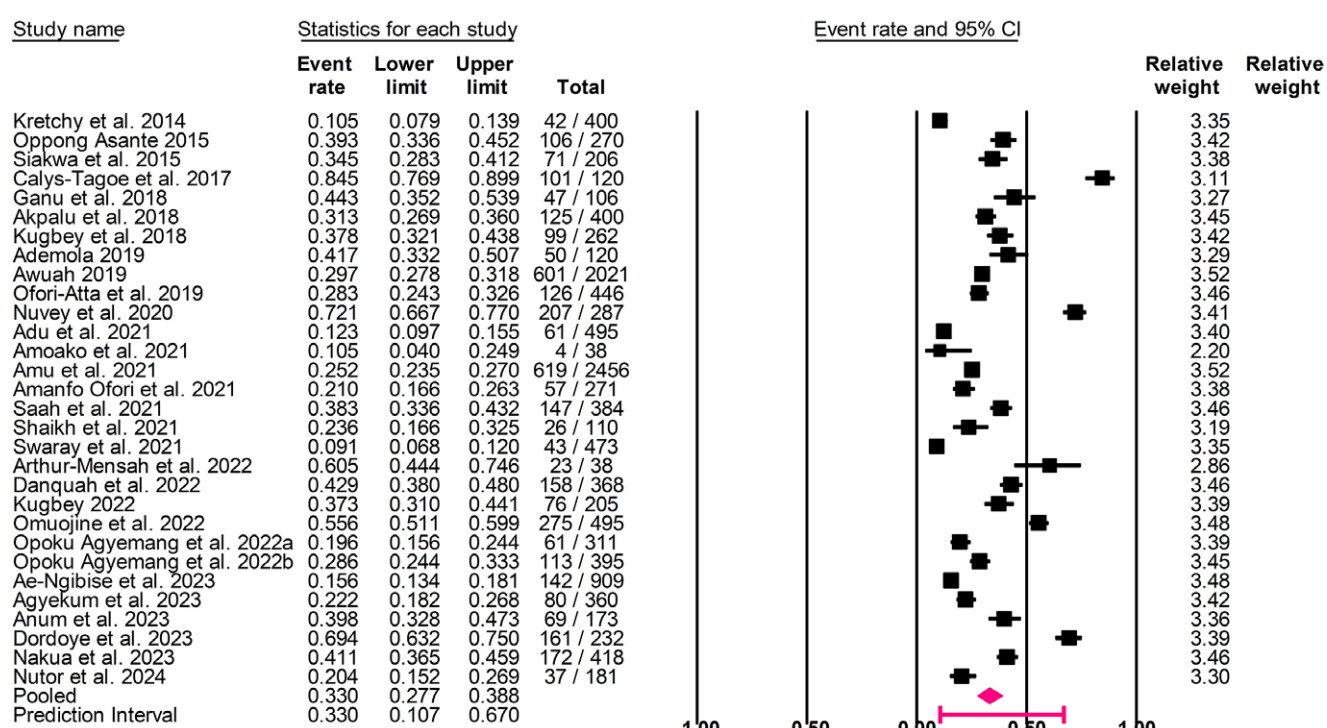

**Figure 3.** Forest plot of the pooled prevalence of depressive disorders and symptoms.

point prevalence of anxiety disorders and symptoms was higher in adults with a comorbid chronic physical health condition (59.0%, 95% CI: 42.3–73.8%, 6 studies) than those without it (32.2%, 95% CI: 23.1–42.9%, 13 studies). Point prevalence of anxiety disorders and symptoms was higher in Greater Accra and Ashanti regions (67.8%, 95% CI: 40.1%–86.8%, 1 study), Northern and Eastern regions (66.0%, 95% CI: 38.1–86.0%, 1 study) and Greater Accra region (60.2%, 95% CI: 46.9–72.2%, 5 studies), compared with Ashanti region (38.8%, 95% CI: 26.3–53.1%, 4 studies) and Central region (35.6%, 95% CI: 19.7–55.5%, 2 studies; see Supplementary Appendix M for the regional distribution of prevalence of anxiety disorders and symptoms). Point prevalence estimate of anxiety disorders and symptoms was higher in studies conducted before 2020 (55.8%, 95% CI: 42.6–68.2%, 6 studies) than those conducted from 2020 onwards (26.1%, 95% CI: 18.7–35.2%, 9 studies).

### Subgroup analysis of the prevalence of depressive disorders and symptoms

Subgroup analysis showed that population type ($Q = 9.426$, $p < 0.05$), region ($Q = 39.912$, $p < 0.05$), sample size ($Q = 7.219$, $p < 0.05$) and time period of data collection ($Q = 12.678$, $p < 0.05$) were significant moderators of depression prevalence (see Supplementary Appendix N). Point prevalence of depressive disorders and symptoms was higher in specific populations (37.7%, 95% CI: 31.6–44.2%, 24 studies) than in the general populations (19.9%, 95% CI: 11.5–32.1%, 4 studies). Point prevalence of depressive disorders and symptoms was higher in Northern and Eastern regions (72.1%, 95% CI: 44.8–89.2%, 1 study) and Greater Accra (46.1%, 95% CI: 36.5–56.0%, 9 studies), and lower in Ashanti and Central regions (10.5%, 95% CI: 2.5–35.3%, 1 study; see

Supplementary Appendix O on regional distribution of prevalence of depressive disorders and symptoms). Point prevalence of depressive disorders and symptoms was higher in studies with small sample size (39.4%, 95% CI: 32.9–47.9%, 18 studies) than those with large sample size (25.0%, 95% CI: 18.7–32.6%, 12 studies). Point prevalence of depressive disorders and symptoms was higher in studies conducted before 2020 (36.9%, 95% CI: 28.1–46.6%, 10 studies) than those conducted from 2020 onwards (22.6%, 95% CI: 16.4–30.2%, 10 studies).

### Social determinants of mental health associated with anxiety symptoms

Seven cross-sectional studies ($N$ = 5,025) reported one or more proximal SDOMH associated with anxiety symptoms (see Table 2). All studies performed multivariable statistical analysis, with adjustment for covariates. None of the studies examined environmental events associated with anxiety symptoms, and SDOMH associated with anxiety disorders.

#### Demographic domain
A total of five studies examined demographic factors that were categorised into three distinct subdomains: age, gender and marital status.

Age. One study found that younger age (*i.e.*, 18–44 years) was associated with higher odds of severe anxiety symptoms (Opoku Agyemang et al., 2022a).

Gender. Gender was examined in four studies, all of which reported consistent results. Among these, two studies found that female gender was significantly associated with increased odds of anxiety symptoms (Saah et al., 2021; Opoku Agyemang et al., 2022a). One study found a small but significant effect showing that females had higher odds of experiencing anxiety symptoms than males (Anum et al., 2023). One study reported that male gender was significantly associated with decreased odds of anxiety symptoms during the COVID-19 pandemic (Boateng et al., 2021).

Marital status. Marital status was examined in two studies, all of which reported consistent results. One study found that living without a regular partner was significantly associated with increased odds of anxiety symptoms (Opoku Agyemang et al., 2022b), whereas the other study found that being married or in a union was significantly associated with lower levels of anxiety symptoms (Boateng et al., 2021).

#### Economic domain
Economic factors were examined in two studies. One study found that high income was significantly associated with lower odds of anxiety symptoms (Amu et al., 2021), whereas the other study reported that anticipating better remuneration was significantly associated with higher odds of anxiety symptoms (Saah et al., 2021).

#### Neighbourhood domain
One study examined this domain and found that living in an urban environment was associated with higher odds of anxiety symptoms compared to living in rural and semi-urban environments (Omuojine et al., 2022).

#### Social and cultural domain
Three studies examined the association between educational attainment and anxiety symptoms. Among these, two studies found that having no formal education was significantly associated with higher levels of anxiety symptoms (Amu et al., 2021; Opoku Agyemang et al., 2022b). One study found that having a master's degree was significantly associated with decreased odds of anxiety symptoms compared to a diploma (Opoku Agyemang et al., 2022a). A study examining the interaction between male gender and education found that males without formal education had higher odds of experiencing anxiety symptoms than those with tertiary education (Opoku Agyemang et al., 2022b).

### Social determinants of mental health associated with depressive symptoms

Twelve cross-sectional studies ($N$ = 6,223) reported one or more proximal SDOMH associated with depressive symptoms (see Table 2). All studies performed multivariable statistical analysis, with adjustment for covariates. None of the studies examined environmental events and neighbourhood factors associated with depressive symptoms and SDOMH associated with depressive disorders.

#### Demographic domain

A total of six cross-sectional studies examined demographic factors that were categorised into five distinct subdomains: age, ethnicity, gender, marital status and religion.

#### Age
One study found that increasing age was significantly associated with higher odds of depressive symptoms, particularly among people living with HIV (Nutor et al., 2024).

#### Ethnicity
The association between ethnicity and depressive symptoms was examined in two studies. One study found that Guans had the lowest odds of experiencing depressive symptoms compared to Akans, Ewes, Mole-Dagbanis and other unspecified ethnic groups (Amu et al., 2021). The other study found that waiters who were Ewe, Ga-Adangme, Mole Dagbani, and identified with other ethnicities had increased odds of experiencing depressive symptoms compared to Akans (Saah et al., 2021).

#### Gender
Gender was examined in four studies, all of which reported consistent results. Among these, two studies found that female gender was significantly associated with increased odds of depressive symptoms (Akpalu et al., 2018; Saah et al., 2021; Opoku Agyemang et al., 2022b). Another study found that men living with HIV had increased odds of experiencing depressive symptoms compared to those without it (Nutor et al., 2024).

#### Marital status
Marital status was examined in two studies, all of which reported consistent results. One study found that, on average, adults who were married had decreased odds of experiencing depressive symptoms compared to those who were unmarried, divorced, widowed or separated (Akpalu et al., 2018). The other study found that being

**Table 2.** Social determinants of mental health associated with anxiety and depressive symptoms among adults in Ghana ($N$ = 11,248, 15 studies)

| Main domain | Sub-category | SDOMH associated with higher odds/ levels of anxiety symptoms | SDOMH associated with lower odds/ levels of anxiety symptoms | SDOMH associated with higher odds/ levels of depressive symptoms | SDOMH associated with lower odds/ levels of depressive symptoms |
|---|---|---|---|---|---|
| Demographic domain | Age | Opoku Agyemang et al. (2022a): Younger age ($\beta$ = −9.68, SE = 128.94, $p < 0.05$)* | x | Nutor et al. (2024): Increasing age ($\beta$ = 0.012, SE = 0.005, 95% CI: 0.002–0.021, $p < 0.05$) | x |
| | Ethnicity | x | x | Saah et al. (2021): Waiters who were Ewe (AOR = 2.68, 95% CI: 0.85–8.50, $p$ = 0.093); Ga-Dangme (AOR = 1.50, 95% CI: 0.57–3.98, $p$ = 0.412); Mole-Dagbani (AOR = 1.05, 95% CI: 0.43–2.55, $p$ = 0.918) and identified with other ethnicities (AOR = 1.02, 95% CI: 0.37–2.78, $p$ = 0.975) | Amu et al. (2021): Being a Guan (AOR = 0.40, 95% CI: 0.23–0.68, $p < 0.05$) |
| | Gender | Anum et al. (2023): Female gender ($B$ = −3.75, 95% CI: −7.50 −0.64, $p < 0.05$)*** Opoku Agyemang et al. (2022a): Female gender ($B$ = −17.516, SE = 0.656, $p < 0.05$) Saah et al. (2021): Female gender (AOR = 1.86; 95% CI: 1.17–2.96, $p$ = 0.009) Boateng et al. (2021): Males living with HIV ($\beta$ = − 0.97, SE = 0.35, $p < 0.01$)* | x | Akpalu et al. (2018): Female gender (OR = 2.84, 95% CI: 1.67–4.07, $p < 0.05$) Opoku Agyemang et al. (2022b): Female gender (AOR = 0.48, 95% CI: 0.25–0.91, $p$ = 0.03) Saah et al. (2021): Female gender (AOR = 1.69; 95% CI: 1.02–2.79, $p$ = 0.041) Nutor et al. (2024): Males living with HIV ($\beta$ = 0.405, SE = 0.16, 95% CI: 0.0079–0.732, $p < 0.05$) | x |
| | Marital status | Opoku Agyemang et al. (2022b): Being without a regular partner (AOR = 0.63, 95% CI: 0.40–1.00: $p$ = 0.049) | Boateng et al. (2021): Being in a union or marital relationship ($\beta$ = −0.77, SE = 0.54, $p < 0.05$)* | Nakua et al. (2023): Being single (AOR = 1.66; 95% CI: 1.08–2.53, $p < 0.05$) | Akpalu et al. (2018): Being married (OR = 1.63, 95% CI: 1.05–2.54, $p < 0.05$) |
| | Religion | x | x | Kugbey et al. (2018): Being a non-Christian (AOR = 5.67, 95% CI: 2.10–15.27, $p < 0.05$) | x |
| Economic domain | Employment | x | x | Adu et al. (2021): Job loss ($\beta$ = 0.30, $p < 0.05$); self-employment ($\beta$ = 1.89, SE = 0.95 95% CI: 0.023–0.977, $p < 0.05$) Amu et al. (2021): no discernible association between occupation and depressive symptoms Saah et al. (2021): Anticipating better remuneration (AOR = 3.09, 95% CI: 1.95–4.87, p < 0.001) | x |
| | Income | Saah et al. (2021): Anticipating better remuneration (AOR = 2.85; 95% CI: 1.82–4.49, $p < 0.001$) | Amu et al. (2021): High income (AOR = 0.34, 95% CI: 0.13–0.86) | Nakua et al. (2023): Less wealth (AOR = 1.97; 95% CI: 1.18–3.27) Ofori-Atta et al. (2019): Low monthly household income ($\beta$ = 4.71, SE = 1.99, $p$ = 0.06) Opoku Agyemang et al. (2022a): Less income (*i.e.*, GH₵1,500.00) ($\beta$ = 18.18, SE = 40.35, $p < 0.05$)* | Amu et al. (2021): High income (AOR = 0.27, 95% CI: 0.11–0.66, $p < 0.05$) |
| Neighbourhood domain | Urban environment | Omuojine et al. (2022): Living in an urban environment (AOR = 1.67, 95% CI: 1.12–2.51, p < 0.05) | x | x | x |
| Environmental domain | | x | x | x | x |
| Social and cultural domain | Traumatic events | x | x | Asante and Andoh-Arthur, 2015: Forced sex (AOR = 9.87, 95% CI: 4.87–20.43, $p < 0.05$), being beaten by an | x |

*(Continued)*

**Table 2.** (*Continued*)

| Main domain | Sub-category | SDOMH associated with higher odds/ levels of anxiety symptoms | SDOMH associated with lower odds/ levels of anxiety symptoms | SDOMH associated with higher odds/ levels of depressive symptoms | SDOMH associated with lower odds/ levels of depressive symptoms |
|---|---|---|---|---|---|
| | | | | intimate partner (AOR = 6.22, 95% CI: 3.10–12.91, $p < 0.05$), childhood physical abuse (AOR = 9.39, 95% CI: 4.72–18.70, $p < 0.05$) and sexual abuse (AOR = 7.23, 95% CI: 3.49–14.99, $p < 0.05$) | |
| | Education | Amu et al. (2021): Having no formal education (AOR = 0.35, 95% CI = 0.17–0.73, $p < 0.05$) Opoku Agyemang et al. (2022b): Having no formal education (AOR = 0.49, 95% CI: 0.30–0.79: $p = 0.004$) Opoku Agyemang et al. (2022b): Males living with HIV without formal education (AOR = 0.18, 95% CI: 0.04–0.86, $p = 0.03$) | Opoku Agyemang et al. (2022a): Having a master's degree ($\beta = -1.85$, SE = 182.85, $p < 0.05$)* | Opoku Agyemang et al. (2022a)**: Holding a master's degree. Opoku Agyemang et al. (2022b): Males living with HIV without formal education (AOR = 0.11, 95% CI: 0.02–0.73, $p = 0.02$) Nutor et al. (2024): Secondary education ($\beta = 0.442$, SE = 0.203, 95% CI: 0.04–0.844, $p < 0.05$) | Amu et al. (2021): Tertiary level of education (AOR = 0.32, 95% CI: 0.15–0.66, $p < 0.05$) Kugbey (2022): Formal education and English reading ability (OR = 0.32*, 95% CI: 0.13–0.82, $p < 0.05$) |
| | Living alone and social support | x | x | Kugbey et al. (2018): Living alone (AOR = 2.36, 95% CI: 1.16–4.83, $p < 0.05$). Nutor et al. (2024): High social support ($\beta = 0.659$, SE = 0.238, 95% CI: 0.187–1.132) Asante and Andoh-Arthur (2015): Lack of social support (OR = 0.96; 95% CI: 0.93–0.99) and AOR = 0.94 (95% CI: 0.91–0.96) | x |

*Note:* X, not reported; AOR, adjusted odds ratio; $\beta$, adjusted linear regression model coefficient; CI, confidence interval; OR, odds ratio; *, 95% confidence interval was not reported; **, uncertainties around measure of effects; ***, standard error not reported.

single was significantly associated with increased odds of depressive symptoms (Nakua et al., 2023).

*Religion*

One study found that being a non-Christian was significantly associated with increased odds of depressive symptoms (Kugbey et al., 2018).

**Economic domain**

Six cross-sectional studies examined the association between economic factors and depressive symptoms. Economic factors were categorised into two sub-domains: employment-related factors and income.

*Employment-related factors*

Employment-related factors were examined in three studies. One study found that job loss and self-employment were significantly associated with increased odds of depressive symptoms (Adu et al., 2021). Another study found that anticipating better remuneration was significantly associated with increased odds of depressive symptoms (Saah et al., 2021). One study found that there was no discernible association between occupation and depressive symptoms (Amu et al., 2021).

*Income*

Income was examined in four studies, all of which reported consistent results. One study found that low income (*i.e.*, ≤ GH₵1,500.00, equivalent to $141.6 USD) was significantly associated with increased odds of depressive symptoms (Opoku Agyemang et al., 2022a). Similarly, another study found that low monthly household income (*i.e.*, GH₵ 50.00, equivalent to 4.71 USD) was associated with higher odds of depressive symptoms; however, only with borderline significance (Ofori-Atta et al., 2019). One study found that high income (> GH₵1,000.00, equivalent to $94.4 USD) was significantly associated with lower odds of depressive symptoms (Amu et al., 2021). One study found that low wealth was significantly associated with increased odds of depressive symptoms (Nakua et al., 2023).

**Social and cultural domain**

Seven cross-sectional studies examined social and cultural factors associated with depressive symptoms. These factors were categorised into three sub-domains: educational attainment, living alone and social support and traumatic experiences.

*Educational attainment*

Educational attainment was variably associated with depressive symptoms across five studies. Among these, one study found that

master's degree holders had increased odds of experiencing severe depressive symptoms compared with diploma and bachelor's degree holders (Opoku Agyemang et al., 2022a). Similarly, another study found that adults with secondary education had increased odds of experiencing depressive symptoms compared to those without formal education (Nutor et al., 2024). However, two studies found that higher educational attainment was significantly associated with decreased odds of depressive symptoms. Among these two studies, one found that adults with tertiary education had decreased odds of experiencing depressive symptoms compared to those without formal education (Amu et al., 2021). The other study found that formal education and English reading ability were significantly associated with decreased odds of depressive symptoms (Kugbey, 2022). One study examining the interaction between gender and education found that males, particularly those living with HIV without formal education, had increased odds of experiencing depressive symptoms compared to those with tertiary education (Opoku Agyemang et al., 2022b).

### Living alone and social support

Two studies found that lack of social support (Asante and Andoh-Arthur, 2015) and high social support (Nutor et al., 2024) were significantly associated with increased odds of depressive symptoms. One study found that living alone was significantly associated with higher odds of depressive symptoms (Kugbey et al., 2018).

### Traumatic experiences

One study found that traumatic experiences, such as being beaten by an intimate partner, childhood physical and sexual abuse, and forced sex, were significantly associated with higher odds of depressive symptoms (Asante and Andoh-Arthur, 2015).

## Discussion

Pooled point prevalence of anxiety disorders and symptoms, depressive disorders and symptoms, and symptoms of psychological distress was 40.3%, 33.0% and 36.7%, respectively, highlighting a significant mental health burden among adults in Ghana. Reviews conducted in other LMICs have reported varying point prevalence rates of anxiety and depression disorders and symptoms among adults (Bello et al., 2022; Kaggwa et al., 2022; Opio et al., 2022). The point prevalence of anxiety disorders (20.2%) and depressive disorders (21.2%) reported in a review from Uganda (Opio et al., 2022) was lower than the estimates observed in our review. Similarly, another review from Uganda reported a slightly lower point prevalence of depressive disorders and symptoms (30.2%) (Kaggwa et al., 2022), compared to our findings. In contrast, a review focusing on the African continent reported higher prevalence rates for anxiety (47%) and depressive symptoms (48%). Potential reasons for discrepancies in prevalence estimates may include the small number of studies on anxiety (5) and depressive disorders (19), and differences in populations studied (*i.e.*, adults and children) in the two reviews from Uganda (Kaggwa et al., 2022; Opio et al., 2022) compared with our review, and the potential impact of the COVID-19 pandemic, which may have elevated prevalence rates across Africa (Bello et al., 2022). Comparisons with other reviews are also limited due to variations in inclusion criteria, population characteristics and methodologies. The high degree of heterogeneity found across studies, predominant use of self-reported screening tools compared to diagnostic interviews, and

high risk of bias identified in quality domains related to sample coverage, sampling methods, condition measurement, and response rate warrant a cautious interpretation of our findings.

Geographical location was a significant moderator of the prevalence of anxiety disorders and symptoms, and prevalence of depressive disorders and symptoms. Higher prevalence rates were reported in studies conducted in the Southern regions of Ghana, particularly Greater Accra and Ashanti region, whereas prevalence studies were limited or not available in the Northern regions of Ghana (*e.g.*, Upper East region). Variability in regional coverage of prevalence data aligns with mental health service distribution in Ghana, which is skewed towards the Southern regions of Ghana, with limited service availability in the Northern regions (Eaton and Ohene, 2016; Mwangi et al., 2023). The Northern regions face socioeconomic challenges attributable to factors including climate change, ethnic conflict, food insecurity, and natural disasters, which may contribute to a higher mental health burden. However, limited research, particularly from the Northern regions of Ghana, limits our understanding of the mental health needs of those residing in these regions (Eaton and Ohene, 2016; Mwangi et al., 2023). We also found that the point prevalence of anxiety disorders and symptoms (54.1%), and the point prevalence of depressive disorders and symptoms (37.7%) were higher in studies conducted before 2020 than those conducted from 2020 onwards, consistent with trends observed in other reviews (Cai et al., 2024). However, some studies reporting the prevalence of anxiety (four studies) and depressive (nine studies) disorders and symptoms did not report time periods for recruitment and data collection, limiting our ability to draw any definitive conclusions about prevalence rates over time. Findings highlight the need to strengthen the reporting practices of prevalence research to enable comparisons of prevalence estimates over time.

Included studies predominantly used self-report screening tools and clinical cut-offs instead of clinician-administered diagnostic interviews. Clinician-administered diagnostic interviews, widely considered the "gold standard" method of assessing anxiety and depressive disorders, often yield lower prevalence estimates than those obtained using clinical cutoffs on self-report screening tools. However, clinician-administered diagnostic interviews tend to be expensive and time-intensive to use for prevalence research or population-wide surveys (Brown et al., 2021). In Ghana, the use of self-report screening tools has been found to be more feasible for large-scale prevalence studies due to the limited number of trained professionals and funding constraints (Agyapong et al., 2015). Yet, self-report screening tools vary in sensitivity and specificity, and may have contributed to higher prevalence estimates reported in our review (Jia et al., 2024). Further, cut-off points used across studies commonly vary (Zimmerman, 2024). In our review, studies using the same self-report screening tool used different cut-offs. For example, studies using the PHQ-9 used cut-offs ranging from ≥3 (three studies) to ≥10 (three studies). The use of culturally adapted and validated self-report screening tools and cut-offs to measure the prevalence of anxiety and depression disorders and symptoms would improve the accuracy and comparability of prevalence estimates in Ghana and across reviews (Deng et al., 2021; Nielsen-Scott et al., 2022).

Subgroup analysis showed that the presence of a chronic physical condition was a significant moderator of the point prevalence of anxiety disorders and symptoms, with higher rates observed among adults with a chronic physical condition (57.3%) than those without (32.2%). This finding is consistent with previous literature

highlighting the disproportionate burden of chronic physical health conditions in LMICs, and their co-occurrence with higher rates of anxiety disorders and symptoms, particularly among adults living with breast cancer (Kyei, 2018; Bello et al., 2022; Rohwer et al., 2023). Higher rates of anxiety disorders and symptoms may pose a significant challenge to the management of chronic physical conditions, interfering with healthcare utilisation, treatment outcomes and quality of life (Kugbey, 2022). Findings suggest a need for regular screening of anxiety disorders and symptoms as part of routine care to inform treatment decisions and appropriate referrals for adults living with chronic physical conditions in Ghana.

Overall, in relation to SDOMH, we found that demographic, economic, social and cultural factors were more frequently studied compared to environmental events and neighbourhood factors. Most evidence was found in the demographic domain, where increasing age, younger age, female gender and marital status were found to be significant predictors of higher odds of anxiety and depressive symptoms in adults, consistent with previous reviews (Silva et al., 2016; Lund et al., 2018). Evidence on social and cultural factors, including childhood physical and sexual abuse, forced sex, intimate partner violence, and educational attainment and their association particularly with depressive symptoms was limited (Asante and Andoh-Arthur, 2015; Kugbey, 2022). The review identified patterns consistent with the social gradient in mental health, whereby job loss, lower wealth, low educational attainment, low income and low social support were associated with increased odds of anxiety and depressive symptoms, consistent with previous research (Alegría et al., 2018; Lund et al., 2018). Some studies found that anticipated increase in remuneration, higher educational attainment, high social support and self-employment were associated with higher odds of depressive symptoms in adults, potentially suggesting that SDOMH may operate through complex and context-dependent pathways that cross-sectional studies may not fully capture (Lund et al., 2018). Importantly, the review found significant gaps regarding neighbourhood factors and environmental events potentially associated with anxiety and depressive symptoms. Research on neighbourhood factors was scarce, with only one study reporting that living in an urban environment was associated with an increased odds of anxiety symptoms (Omuojine et al., 2022). This finding is consistent with previous research reporting that urban living exposes residents to anxiogenic factors, such as high stress levels, noise pollution, violence and reduced social cohesion, compared to rural and semi-urban environments (Silva et al., 2016; Lund et al., 2018). None of the reviewed studies examined the association between environmental events, including climate change, forced migration, natural disasters and anxiety and depressive disorders and symptoms. Findings should be interpreted cautiously, given that few studies explored the same SDOMH, and our understanding of both the interaction between SDOMH and how SDOMH contributes to the varying burden of anxiety and depressive symptoms is limited.

## Strengths and limitations

This review has several strengths. We utilised a comprehensive electronic search strategy, peer reviewed following the PRESS guideline to minimise the risk of missing relevant studies. We adhered to quality standards informed by the JBI critical appraisal checklist and PRISMA checklist, with screening steps conducted by two independent reviewers. We included studies of chronic physical health populations, recognising that mental health difficulties

often co-occur with physical health conditions. We explored SDOMH associated with anxiety and depressive disorders and symptoms, which are relevant for achieving the SDGs (*e.g.*, SDG 1 [ending poverty], SDG 5 [gender equality], SDG 8 [decent work and economic growth], SDG 9 [industry, innovation and infrastructure], and SDG 10 [reducing inequalities], and informing strategies to reduce the mental health burden among adults.

Limitations of the review included substantial heterogeneity arising from variations in study population and assessment methods, limiting direct comparison with other reviews. While sources of heterogeneity were examined *via* subgroup analysis, other potential moderators of prevalence estimate, for example, mean age and income, were not examined *via* meta-regression due to inconsistent reporting. Most studies used different self-reported screening tools with varied cut-offs rather than diagnostic interviews to measure the prevalence of anxiety and depressive disorders, which may have skewed prevalence estimates. Another major limitation of the review is the over-reliance of studies on cross-sectional designs (100%), limiting causal inference and theorisation of the direction and importance of relative SDOMH over time. Most of the reviewed studies examined SDOMH as isolated factors without examining the complex interactions among SDOMH, including how proximal SDOMH interact with distal SDOMH, or specifying pathways through which SDOMH influence mental health outcomes. Qualitative studies were excluded, limiting an in-depth exploration of the perspectives of adults in Ghana on SDOMH potentially associated with anxiety and depressive disorders and symptoms, which may have provided important insights to inform future intervention development.

## Implications for clinical practice and policy

Current evidence, although limited and largely cross-sectional, suggests a potential association between proximal SDOMH and anxiety and depressive symptoms in adults. Addressing the adverse impacts of SDOMH and increasing access to mental health services *via* the NHIS can facilitate early diagnosis and treatment, and reduce health inequities. A range of intervention study designs (*e.g.*, randomised controlled trials) are required to examine the effect of individual-level and population-level interventions, such as awareness campaigns, cash transfers (*e.g.*, Livelihood Empowerment Against Poverty (LEAP programme), neighbourhood regeneration programmes, social prescribing, trauma-informed therapies and vocational support, on adult mental health outcomes (Kirkbride et al., 2024). Importantly, Ghana's Mental Health Policy 2019–2030 (Government of Ghana, 2018) makes little to no explicit reference to SDOMH, with socioeconomic factors mentioned only once. There is an urgent need for policymakers in Ghana to revise existing mental health policies and programmes to explicitly incorporate SDOMH considerations and allocate adequate financial resources for their implementation. In alignment with resolution WHA74.16 (World Health Organization, 2024), which calls on member states to monitor and act on SDOMH. It is crucial that the Government of Ghana develops robust indicators to track SDOMH and the mental health status of diverse populations in Ghana. Further, clinical monitoring, documentation and interventions addressing SDOMH are gaining international acceptance (Andermann, 2018), and more research is required to evaluate the impact, acceptability, effectiveness and feasibility of these practices within the Ghanaian health context.

## Implications for research

Over-reliance on self-report screening tools that have been validated for adult populations in other contexts, without adaptation to the Ghanaian context, risks overlooking culturally specific expressions of symptoms of anxiety, depression and distress (*e.g.*, brain fatigue and tension in the head) (Ali et al., 2016; Desai and Chaturvedi, 2017). Further, globally recommended cut-offs may not be applicable to LMIC populations, and their use without adaptation to the local context may lead to inaccuracies in prevalence estimates (Ali et al., 2016). Therefore, future research should prioritise the cultural adaptation and validation of clinician-administered diagnostic interviews and self-report screening tools to ensure accurate assessment of anxiety and depressive symptoms in adult populations in Ghana.

We assessed publication bias despite contention around its appropriateness for systematic reviews of prevalence, that is, as prevalence studies do not test for statistical significance, the notion of publication bias (*e.g.*, statistically significant studies are more likely to be published) may not be applicable to prevalence studies (Peters et al., 2006; Borenstein, 2019). Future research should investigate the advantages and disadvantages of assessing publication bias in prevalence studies, as well as consider other sources of potential bias (*e.g.*, omitted variable bias and Neyman bias), which may be particularly relevant to prevalence research (Tripepi et al., 2008). Future studies should prioritise longitudinal, multilevel and integrative study designs that can effectively capture the complex interactions between proximal and distal SDOMH, and their relative importance over time.

## Conclusions

Anxiety and depressive symptoms are common among adults in Ghana. Social determinants of mental health, including educational attainment, female gender, low income, urban environment and traumatic experiences, may be associated with the prevalence of anxiety and depressive symptoms. While our findings indicate a potentially significant mental health burden in Ghana, high levels of heterogeneity and over-reliance on self-reported screening tools suggest a need for caution in interpreting prevalence estimates. Research training programs and financial resources are needed to support the conduct of high-quality prevalence research that can guide clinical interventions and policy planning.

**Open peer review.** To view the open peer review materials for this article, please visit http://doi.org/10.1017/gmh.2025.10122.

**Supplementary material.** The supplementary material for this article can be found at http://doi.org/10.1017/gmh.2025.10122.

**Data availability statement.** All data are provided in the article and Supplementary Material.

**Acknowledgements.** The authors would like to thank Mattias Axén and Malin Barkelind, librarians at Uppsala University Library, for assisting with the electronic search strategy. The authors would also like to thank Alkistis Skalkidou at Uppsala University and Lene Lindberg at Karolinska Institute for peer reviewing the search strategy. We would like to thank Chelsea Coumoundouros for her support during the initial stages of this review.

**Author contribution.** Victoria Awortwe: Conceptualisation; methodology; validation; formal analysis; investigation; data curation; writing – original draft; writing – review and editing; visualisation; project administration. Febrina Maharani: Validation; formal analysis; investigation; data curation; writing – review and editing; visualisation. Meena Daivadanam: Conceptualisation; methodology; writing – review and editing; project administration; supervision. Samuel Adjorlolo: Methodology; writing – review and editing; supervision. Erik M. G. Olsson: Methodology; writing – review and editing; supervision. Louise von-Essen: Resources; writing – review and editing; funding acquisition. Vian Rajabzadeh: Validation; investigation; writing – review and editing. Joanne Woodford: Conceptualisation; methodology; validation; investigation; writing – review and editing; project administration; supervision.

**Financial support.** This work was supported by U-CARE, which is a strategic research environment funded by the Swedish Research Council (dnr 2009–1,093). This funding source had no role in the design of this review, its execution, analyses, interpretation or decision to submit results.

**Competing interests.** The authors declare none.

**Ethics statement.** No ethical approval was required as this review synthesised only publicly available data.

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
