## [Reviewer Report]

Thank you for the opportunity to review your manuscript titled “Prevalence and social determinants of anxiety and depressive disorders and symptoms among adults in Ghana: a systematic review and meta-analysis.”

While I appreciate the relevance of the topic and the methodological approach taken, I have serious concerns regarding the transparency of the included studies in your meta-analysis. Specifically, several studies cited in your quantitative synthesis are not listed in the reference section. It is possible that these are included in the appendices, but I currently do not have access to them.

I previously contacted the editorial office to obtain the appendices so I could review the full list of included studies, but unfortunately, I have not received a response. Without access to these sources, I am unable to verify the inclusion criteria or assess the reliability and appropriateness of the data used in the meta-analysis.

Given these limitations and the absence of crucial supporting documentation, I am unable to conduct a full and accurate assessment. Therefore, I must recommend rejection of the manuscript at this time.

Kind regards,

---

## [Reviewer Report]

This is an impactful and methodologically sound systematic review. I appreciate the authors for undertaking this work and would suggest a few minor changes to bring more clarity to the work and contextualize the same for future research and healthcare decision-making.

First, in the background, the authors need to provide broader health systems scenario and current state of disease burden in Ghana, then present recent epidemiological evidence on mental disorders- which may provide more “context” for this work, and possible enhance the justification of or the need for this review.

Second, the authors mentioned they “utilized a peer-reviewed” strategy (first sentence in the strengths/limitations), and they used gray literature search within the method. It is not clear if they included “peer-reviewed papers” only. Please clarify the inclusion/exclusion of non-peer reviewed documents in this review.

Third, the presentation of social determinants of anxiety and depression needs to be elaborated. Instead of presenting them together, it might be beneficial to present them separately- and later in the discussion, they can be discussed/examined together to see shared risk and protective factors.

Fourth, the correlates of anxiety and depression appears to be summarized rather than synthesized, meaning an in-depth analysis of how risk factors are distributed in the populations, how to interact with each other, or how they contribute to varying burden of anxiety and/or depression- can enrich the discussion. Possibly, this synthesis can lead to theorizing mental health crises in Ghana within the current socio-demographic and economic conditions.

Fifth, the authors may wish to reflect on existing interventions, program, policies, and services related to mental health in Ghana, and contextualize the current evidence to provide a more focused set of research and policy recommendations that may inform how current gaps can be addressed and how the evidence provided in this paper can facilitate the same.

---

## [Editor Report]

May you kindly address the queries from the reviewers, in particular, ensuring that all supporting data is provided to ensure transparency, methodological rigour and replicability.

---

## [Editor Report]

The outstanding comments and recommendations from our reviewers have been well addressed. This manuscript can now be considered for publication